# Conjuring Semantic Similarity

## Abstract

The semantic similarity between sample expressions measures the distance between their latent 'meaning'. Such meanings are themselves typically represented by textual expressions, often insufficient to differentiate concepts at fine granularity. We propose a novel approach whereby the semantic similarity among textual expressions is based *not* on other expressions they can be rephrased as, but rather based on the *imagery* they evoke. While this is not possible with humans, generative models allow us to easily visualize and compare generated images, or their distribution, evoked by a textual prompt. Therefore, we characterize the semantic similarity between two textual expressions simply as the distance between image distributions they induce, or 'conjure.' We show that by choosing the Jensen-Shannon divergence between the reverse-time diffusion stochastic differential equations (SDEs) induced by each textual expression, this can be directly computed via Monte-Carlo sampling. Our method contributes a novel perspective on semantic similarity that not only aligns with human-annotated scores, but also opens up new avenues for the evaluation of text-conditioned generative models while offering better interpretability of their learnt representations.

## 1 Introduction

Semantic similarity is about comparing data not directly, but based on their underlying 'concepts' or 'meanings'. Since meanings are most commonly expressed through natural language, various methods have attempted to compute them in this space. Words have been often compared based on the occurrences of other words that surround them, and images have likewise been compared 'semantically' by using the text captions that describe them.

While measuring semantic similarity comes natural to humans who share significant knowledge and experience, defining semantic similarity for trained models is non-trivial since text can often be ambiguous or open to multiple (subjective) interpretations. On the other hand, visual elements often transcend communication barriers and allow for comparison at finer granularity. Comparing images is also relatively simple – unlike words, pixel values do not depend on distant knowledge or context.

Hence, instead of comparing images by the captions which describe them, we propose the converse: comparing textual expressions in terms of the images they conjure. In other words, we propose an expanded notion of meaning that is purely "visually-grounded". This would be hard if not impossible for humans, since the process requires visualizing and comparing 'mental images' each individual can conceive, but it is straightforward for trained models.

We propose to leverage modern image generative models, in particular the class of text-conditioned diffusion models, for this purpose. By doing so, the semantic similarity between two text passages can simply be measured by the similarity of image distributions generated by a model conditioned on those passages.

There is a technical nugget that needs to be developed for the method to be viable, which is how to compare diffusions in the space of images. To address this, we propose to leverage the Jensen Shannon Divergence between the stochastic differential equations (SDEs) that govern the flow of the diffusion model, which we will show to be computable using a Monte-Carlo sampling approach.

We will show that our simple choice of distance already leads to results comparable to zero-shot approaches based on large language models. To validate our proposed definitions, we further run ab-

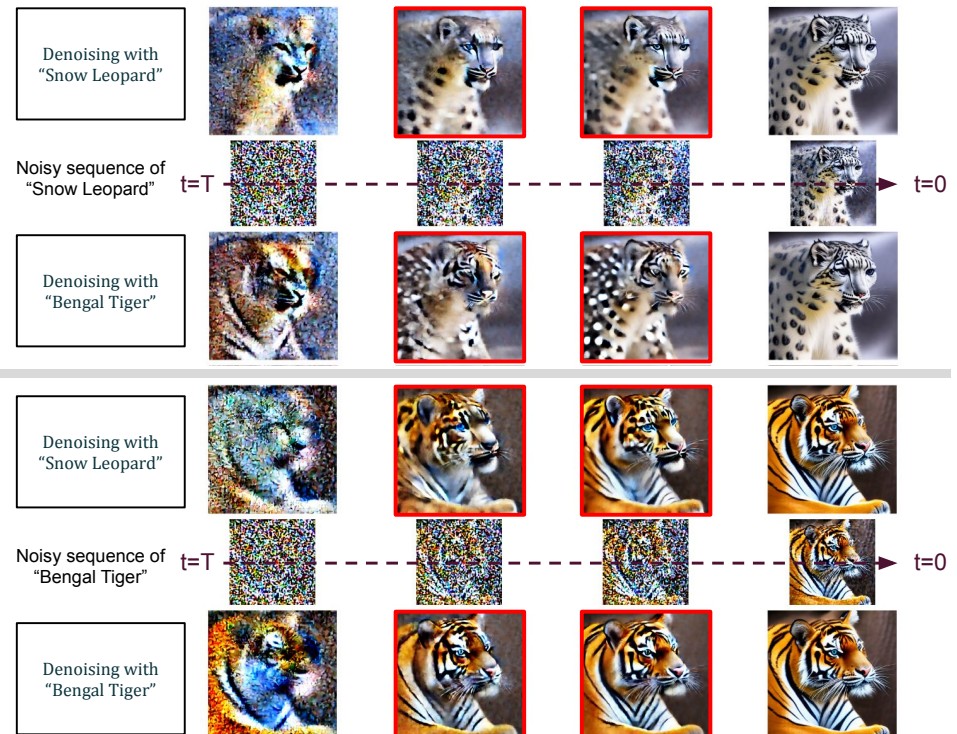

Figure 1: We illustrate the process of conjuring semantic similarity between textual expressions "Snow Leopard" and "Bengal Tiger". We denoise each sequence of noisy images (middle row of both halves of figure) with both prompts (top and bottom row of both halves of figure). Our method can be interpreted as taking the Euclidean distance between the resulting images in the two rows. The sequences of noisy images are obtained with either of the two text expressions (top / bottom halves of Figure) starting from a Gaussian prior ($t = T$). Observing cells highlighted in red, we see that the model converts pictures of Snow Leopards into Bengal Tigers by changing their characteristic spotted coats into stripes, and adding striped textures to the animal's face (top half of Figure), and conversely converts Bengal Tigers into Snow Leopards by changing their characteristic stripes into spotted coats (bottom half of Figure). This enables interpretability of their semantic differences via changes in their evoked imageries.

lation studies over several components of our method, demonstrating robustness to specific choices of diffusion models and their inference algorithms.

To summarize the contributions of this work, we propose an approach for evaluating semantic similarity between text expressions that is grounded in the space of visual images. Our method has a unique advantage over traditional language-based methods that, in addition to providing a numerical score, it also provides a visual expression, or 'explanation', for comparison, enabling better interpretability of the learnt representations (Figure 1). Additionally, our method is the first to enable quantifying the alignment of semantic representations learnt by diffusion models compared to that of humans, which can open up new avenues for the evaluation text-conditioned diffusion models. Finally, we note that our approach can be used to derive many possible variants based on the metric used to compare images, which we leave for future exploration.

## 2    RELATED WORKS

**Text-Conditioned Image Generative Models.**    We begin by briefly surveying the literature on text-conditioned image generation. Goodfellow et al. (2020) proposed the Generative Adversarial Network (GAN), a deep learning-based approach for image generation. Mirza & Osindero (2014) proposed a method for conditioning GANs based on specified labels. Works such as VQ-VAE

(Van Den Oord et al., 2017) and VQ-VAE-2 (Razavi et al., 2019) also built upon the foundational works of Variational Auto Encoders (VAEs) (Kingma, 2013) to learn discrete representations used to generate high quality images, among other outputs, when paired with an autoregressive prior. DALL-E (Ramesh et al., 2021) has also been developed as an effective text-to-image generator leveraging autoregressive Transformers (Vaswani, 2017).

Our work focuses on diffusion models (Sohl-Dickstein et al., 2015), which have achieved state-of-the-art results among modern image generation models. These models can be viewed from the perspective of score-based generative models (Song et al., 2020b), which represent the distribution of data via gradients, and are sampled from using Langevin dynamics (Welling & Teh, 2011). Diffusion models have been trained by optimizing the variational bound on data likelihood (Ho et al., 2020). Modeled as stochastic differential equations (Song et al., 2020b), the same training objective has also been shown to enable maximum-likelihood training under specific weighing schemes (Song et al., 2021). Diffusion models have also demonstrated strong results on conditional image generation tasks. Sohl-Dickstein et al. (2015) and Song et al. (2020b) showed that gradients of a classifier can be used to condition a pre-trained diffusion model. Dhariwal & Nichol (2021) introduced Classifier-guidance, which achieved state-of-the-art results in image synthesis at the time it was released. Song et al. (2020a) introduced Denoising Diffusion Implicit Models (DDIM), greatly accelerating the process of sampling from trained diffusion models as compared to DDPM (Ho et al., 2020). Karras et al. (2022) presented a unified view of existing diffusion models from a practical standpoint, enabling develop improved sampling and training techniques to obtain greatly improved results.

**Semantic Space of Generative Models.** The Distributional Hypothesis (Harris, 1954) forms the basis for statistical semantics, characterizing the meaning of linguistic items based on their usage distributions. This is also closely related to Wittgenstein's use theory of meaning (Wittgenstein, 1953), often popularized as "meaning is use". Many methods have been developed in machine learning and Natural Language Processing (NLP) literature to compute these semantic spaces, including Word2Vec (Mikolov et al., 2013). In light of modern Large Language Models (LLMs), Liu et al. (2023) defined the space of meanings for autoregressive models to be the distribution over model continuations for any input sequence. This has been used to define notions of semantic distances and semantic containment between textual inputs. Achille et al. (2024) further defined conceptual similarity between images by projecting them into the space of distributions over complexity-constrained captions, producing similarity scores that strongly correlate with human annotations. Soatto et al. (2023) generalizes these definitions by considering meanings as equivalence classes, where partitions can be induced by either an external agent or the model itself. Vector-based representations such as those obtained from CLIP Radford et al. (2021), or in general any sentence embedding model (Devlin et al., 2018; Opitz & Frank, 2022) have also been defined specifically for the computation of semantic distances. Such representations, however, are often difficult to interpret.

In contrast to these works, Bender & Koller (2020) argues that training on language alone is insufficient to capture semantics, which they argue requires a notion of "communicative intents" that are external to language. In this paper, we explore a notion of meaning that is grounded in the distribution of evoked imageries, and present a simple algorithm to compute interpretable distances in this space for the class of text-conditioned diffusion models.

**Evaluation and Interpretation of Diffusion Models.** Common metrics used to evaluate diffusion models are, among many others, the widely-used FID (Heusel et al., 2017) score, Kernel Inception Distance (Bińkowski et al., 2018), and the CLIP score (Hessel et al., 2021). Despite these choices, Stein et al. (2024) recently discovered that no existing metric used to evaluate diffusion models correlates strongly with human evaluations. While existing evaluations often focus on the quality and diversity of generations, our method is the first to evaluate semantic alignment of the representations learnt by diffusion models. Several techniques have also been developed to interpret the generation of diffusion models. Kwon et al. (2022); Park et al. (2023) edit the bottleneck representations within the U-Net architecture of diffusion models for semantic image manipulation. Gandikota et al. (2023) identifies low-rank directions corresponding to various semantic concepts, and Li et al. (2023) shows that diffusion models can be used for image classification. Kong et al. (2023a;b) frame diffusion models using information theory, improving interpretability of their learnt semantic relations. Orthogonal to these works, our method enables visualizing the semantic relations be-

---

**Algorithm 1** Conjuring Semantic Similarity

---

**Require:** Diffusion model $s_\theta$, Prompts $y_1, y_2$, Monte-Carlo steps $k$
   Initialize $d = 0$
   **for** $i = 1 \dots k$ **do**
      $x_T \leftarrow$ Sample from initial distribution $\pi$
      $\hat{x}_T, \dots, \hat{x}_0 \leftarrow$ Denoise $x_T$ conditioned on $y_1$
      $\tilde{x}_T, \dots, \tilde{x}_0 \leftarrow$ Denoise $x_T$ conditioned on $y_2$
      $d \leftarrow d + \frac{1}{T} \sum_{t=1}^{T} \|s_\theta(\hat{x}_t, t | y_1) - s_\theta(\hat{x}_t, t | y_2)\|_2^2$
      $d \leftarrow d + \frac{1}{T} \sum_{t=1}^{T} \|s_\theta(\tilde{x}_t, t | y_1) - s_\theta(\tilde{x}_t, t | y_2)\|_2^2$
   **end for**
   **return** $d/k$                            ▷ Return similarity score

---

tween textual prompts in natural language learnt by diffusion models via the distributions over their generated imageries.

## 3 METHOD

We will present a short preliminary on (conditional) diffusion models in Section 3.1, and derive our algorithm for computing semantic similarity in Section 3.2.

### 3.1 PRELIMINARY

Our derivations will leverage Song et al. (2020b)'s the SDE formulation of diffusion models when viewed from the lens of score-based generaive modeling. In particular, a (forward) diffusion process $\{\boldsymbol{x}(t)\}_{t=0}^{T}$ can be modeled as the solution to the following SDE:

$$d\boldsymbol{x} = \boldsymbol{f}(\boldsymbol{x}, t)dt + g(t)d\boldsymbol{w}_t$$

with drift coefficient $\boldsymbol{f} : \mathbb{R}^d \times \mathbb{R} \mapsto \mathbb{R}^d$ and (scalar) diffusion coefficient $g : \mathbb{R} \mapsto \mathbb{R}$, where $\boldsymbol{w}_t$ is standard Brownian motion. We constrain the timesteps $t$ such that $t \in [0, T]$, where $\boldsymbol{x}(0)$ represents the distribution of "fully-denoised" images generated by the diffusion model. We also assume that by construction, the prior at time $T$ is known and distributed according to $\boldsymbol{x}(T) \sim \pi$, where often $\pi = N(0, I)$.

Once trained, we can view text-conditioned diffusion models as a map $s_\theta(\boldsymbol{x}, t | y)$ parameterized by $\theta$ and conditioned on a textual prompt $y \in \mathcal{Y}$ that is used to approximate the score function (Song et al., 2020b), where $s_\theta(\cdot, t | y) : \mathbb{R}^d \mapsto \mathbb{R}^d$ and $\mathcal{Y}$ is the set of textual expressions. As such, each conditional model $s_\theta(\boldsymbol{x}, t | y)$ defines a reverse-time SDE given by:

$$d\boldsymbol{x} = [f(\boldsymbol{x}, t) - g(t)^2 s_\theta(\boldsymbol{x}, t | y)]dt + g(t)d\bar{\boldsymbol{w}}_t \tag{1}$$

where $\bar{\boldsymbol{w}}_t$ is the Brownian motion running backwards in time from $t = T$ to $t = 0$. For ease of notation, we will denote $\mu_\theta(\boldsymbol{x}, t, y) := [f(\boldsymbol{x}, t) - g(t)^2 s_\theta(\boldsymbol{x}, t | y)]$.

### 3.2 CONSTRUCTION

Given two textual prompts $y_1$ and $y_2$, we obtain two separate diffusion SDEs in the space of images using eq. (1), which are given by

$$d\boldsymbol{x}_1 = \mu_\theta(\boldsymbol{x}_1, t, y_1)dt + g(t)d\bar{\boldsymbol{w}}_t \tag{2}$$

$$d\boldsymbol{x}_2 = \mu_\theta(\boldsymbol{x}_2, t, y_2)dt + g(t)d\bar{\boldsymbol{w}}_t \tag{3}$$

We assume the standard conditions for existence and uniqueness of their solutions, in particular for all $t \in [0, T]$ we have $\mathbb{E}\left[\|\boldsymbol{x}(t)\|_2^2\right] \leq \infty$, and for all $y \in \mathcal{Y}$ and $x, x' \in \mathbb{R}^d$, there exists constants $C, D$ such that $\|\mu_\theta(x, t, y)\|_2 + \|g(t)\|_2 \leq C(1 + \|x\|_2)$ and $\|\mu_\theta(x, t, y) - \mu_\theta(x', t, y)\|_2 \leq D\|x - x'\|_2$. We further assume Novikov's Condition holds for all pairs $y_1, y_2 \in \mathcal{Y}$, in particular we are guaranteed the following: $\mathbb{E}\left[\exp\left(\frac{1}{2}\int_0^T \|\mu_\theta(\boldsymbol{x}, t, y_2) - \mu_\theta(\boldsymbol{x}, t, y_1)\|_2^2 dt\right)\right] < \infty$.

Since our goal is to define a semantic distance between textual prompts $y_1$ and $y_2$ by comparing the distributions over images that they conjure, we can achieve this by computing a discrepancy function between the respective SDEs that they induce. In particular, we will use the Jensen–Shannon (JS) divergence, which is simply the symmetrized Kullback–Leibler (KL) divergence between two SDEs. In the following, we show how this divergence can be computed via a Monte-Carlo approach.

Denote the path measures associated with eq. (2) and eq. (3) respectively to be $\mathbb{P}_1$ and $\mathbb{P}_2$. Then, the KL divergence between the two SDEs are defined via

$$D_{KL}(\mathbb{P}_2||\mathbb{P}_1) = -\mathbb{E}_{\mathbb{P}_2} \log\left(\frac{d\mathbb{P}_1}{d\mathbb{P}_2}\right)$$

where $\frac{d\mathbb{P}_1}{d\mathbb{P}_2}$ is the Radon-Nikodym derivative. By Girsanov theorem (Girsanov, 1960), we can compute this derivative as the stochastic exponential given by

$$\exp\left(\int_0^T -\frac{1}{g(t)}\left(\mu_\theta(\boldsymbol{x}, t, y_1) - \mu_\theta(\boldsymbol{x}, t, y_2)\right) d\bar{\boldsymbol{w}}_t - \frac{1}{2}\int_0^T \frac{1}{g(t)^2}\|\mu_\theta(\boldsymbol{x}, t, y_1) - \mu_\theta(\boldsymbol{x}, t, y_2)\|_2^2 dt\right)$$

Novikov's Condition guarantees that this term is a Martingale so, following the approach of Song et al. (2021), the KL divergence between the two SDEs can be simplified as

$$D_{KL}(\mathbb{P}_2||\mathbb{P}_1) = \frac{1}{2}\mathbb{E}_{\mathbb{P}_2}\left[\int_0^T \frac{1}{g(t)^2}\|\mu_\theta(\boldsymbol{x}, t, y_1) - \mu_\theta(\boldsymbol{x}, t, y_2)\|_2^2 dt\right]$$

$$= \frac{1}{2}\mathbb{E}_{\mathbb{P}_2}\left[\int_0^T g(t)^2\|s_\theta(\boldsymbol{x}, t|y_1) - s_\theta(\boldsymbol{x}, t|y_2)\|_2^2 dt\right]$$

We can symmetrize to form our desired distance function:

$$d_{ours}(y_1, y_2) := JSM(\mathbb{P}_1||\mathbb{P}_2) = \frac{1}{2}D_{KL}(\mathbb{P}_2||\mathbb{P}_1) + \frac{1}{2}D_{KL}(\mathbb{P}_1||\mathbb{P}_2)$$

which, ignoring constants, can be written as

$$d_{ours}(y_1, y_2) = \mathbb{E}_{t\sim\text{unif}([0,T]), \boldsymbol{x}\sim\frac{1}{2}p_t(\boldsymbol{x}|y_1)+\frac{1}{2}p_t(\boldsymbol{x}|y_2)}\left[g(t)^2\|s_\theta(\boldsymbol{x}, t|y_1) - s_\theta(\boldsymbol{x}, t|y_2)\|_2^2\right]$$

where $p_t(\cdot|y)$ is the distribution of noisy images at timestep $t$. Similar to how losses at different timesteps are weighted uniformly in the training of real-world diffusion models (*e.g.* $L_{simple}$ proposed by Ho et al. (2020)), we set $g(t)$ to be constant, in particular $g(t) = 1$, to simplify our algorithm such that it does not have to be tailored specifically to each choice of scheduler.

We can compute this resulting semantic distance using Monte-Carlo by discretizing the timesteps to $t \in \{1, \ldots, T\}$. In particular, this is computed in practice via sampling an initial noise vector $\boldsymbol{x}(T) \sim \pi$, and denoising it with both $y_1$ and $y_2$ to obtain a sequence of samples $x_t$'s, and computing the difference in predictions $\|s_\theta(x_t, t|y_1) - s_\theta(x_t, t|y_2)\|_2^2$ at each denoising timestep. We describe this process in Algorithm 1.

## 4 EXPERIMENTS

We describe implementation details in Section 4.1, empirical validation for our definitions in Section 4.3, and ablations in Section 4.4.

### 4.1 IMPLEMENTATION DETAILS

We use Stable Diffusion v1.4 (Rombach et al., 2022), a text-conditioned diffusion model, for all our experiments. For sampling, we use classifier-free guidance (Ho & Salimans, 2022) with guidance scale of 7.5, and sample using the LMS Scheduler (Karras et al., 2022). We specify image sizes to be $512 \times 512$, but note that Stable Diffusion v1.4 uses latent diffusion, as such model predictions are in practice of dimension $64 \times 64$. We compute the Euclidean distance directly in this space. However, for visualization experiments such as in Figure 1, we decode the predictions using the VAE before plotting them. We set $T = 10$ in our experiments to ensure computational feasibility, after ablating

over other choices in Table 3. We perform all our experiments with a single RTX 4090 GPU, and each Monte-Carlo step takes around 2.0s to complete in our naive implementation.

As a technicality, while we model the denoising direction term specified in the reverse-time SDE in eq. (1) as the (text-conditioned) model output $s_\theta(\cdot|y)$, the exact implementation varies depending on the text-conditioning method used. However, we note that in the case of classifier-free guidance, the resulting distance computed using the model output as $s_\theta(\cdot|y)$ is equivalent to that computed using classifier-guidance directions up to proportionality.

### 4.2 BASELINES

While there exists no comparable baselines for quantifying semantic similarity in text-conditioned diffusion models at the time of writing, we present several derivatives of our method below which are directly comparable against:

**Prediction at initial timestep:** We compare the one-step predicted noise vector at the initial denoising timestep. This is defined as $d_{\text{initial}} := \mathbb{E}_{\mathbf{x}\sim\pi}\left[\|s_\theta(\boldsymbol{x}, T|y_1) - s_\theta(\boldsymbol{x}, T|y_2)\|_2^2\right]$.

**Prediction at final timestep:** We compare the predicted noise vector at the final denoising timestep. This is defined as $d_{\text{final}} := \mathbb{E}_{\boldsymbol{x}\sim\frac{1}{2}p_0(\boldsymbol{x}|y_1)+\frac{1}{2}p_0(\boldsymbol{x}|y_2)}\left[\|s_\theta(\boldsymbol{x}, 0|y_1) - s_\theta(\boldsymbol{x}, 0|y_2)\|_2^2\right]$.

**Direct output comparisons:** For the same initial condition, we directly compute the difference between the images produced by the two different labels. This is defined as $d_{\text{output}} := \mathbb{E}_{\boldsymbol{x}\sim\pi}\left[\|\psi_\theta(\boldsymbol{x}|y_1) - \psi_\theta(\boldsymbol{x}, |y_2)\|_2^2\right]$, where $\psi_\theta(\cdot|y) : \mathbb{R}^d \mapsto \mathbb{R}^d$ represents the full reverse diffusion process from the noise prior $\pi$ at time $T$ to the output distribution at time 0.

**KL-Divergence:** We evaluate the non-symmetrized version of our method, computed via the KL divergence between the SDEs obtained from different prompts: $d_{\text{ours-KL}}(y_1, y_2) := \mathbb{E}_{t\sim\text{unif}([0,T]), \boldsymbol{x}\sim p_t(\boldsymbol{x}|y_1)}\left[g(t)^2\|s_\theta(\boldsymbol{x}, t|y_1) - s_\theta(\boldsymbol{x}, t|y_2)\|_2^2\right]$

Using the same parameters as our proposed method, we implement all baselines via Monte-Carlo sampling.

### 4.3 EMPIRICAL VALIDATION

While our work defines a notion of semantic distance grounded in evoked imagery, the validity of this definition hinges on its use as a measure of similarity that aligns with humans'. In particular, we should expect our definition to produce measurements of similarity that agree often with human annotators (which can be viewed as "ground-truth").

To quantify this, we use the Semantic Textual Similarity (STS) (Agirre et al., 2012; 2013; 2014; 2015; 2016; Cer et al., 2017) and Sentences Involving Compositional Knowledge (SICK-R) (Marelli et al., 2014) datasets, containing pairs of sentences each labelled by human annotators with a semantic similarity score ranging from 0-5. We then use our method to compute the image-grounded similarity score, and measure their resulting Spearman Correlation with the annotations.

Interestingly, our experiments in Table 1 show that our visually-grounded similarity scores exhibit significant degrees of correlation with that annotated by humans. While, expectedly, our method presently lags behind embedding models trained specifically for semantic comparison tasks, we show that our visually-grounded similarity scores can rival that produced by existing large language models up to 33B in size. Since our work is the first to formalize and evaluate semantic alignment for this class of conditioned diffusion models, we have also included results from baselines with which our approach is directly comparable against. Our method convincingly outperforms all baseline methods for evaluating semantic similarity in diffusion models. As a remark, we note the large standard deviations in these scores across datasets for all methods, suggesting that what matters for validation purposes is that our distances correlate significantly with that of annotators, rather than obtaining "state-of-the-art" alignment scores since 'ground truth' is fundamentally subjective in each of these benchmarks.

We further remark that in most existing text-conditioned diffusion models, the representation structures that can be captured by our method are limited by those learnt by text-encoder models such as CLIP (Radford et al., 2021), since these encoders are often used to pre-process textual prompts.

Table 1: Comparison with zero-shot methods on Semantic Textual Similarity benchmarks. $*; \dagger; \ddagger$ indicate results taken from Ni et al. (2021); Gao et al. (2021); Liu et al. (2023) respectively. Expectedly, our zero-shot approach does not perform as well as embedding models such as CLIP (Radford et al., 2021) and SimCSE-BERT (Gao et al., 2021), which are trained specifically for semantic comparison tasks. Nevertheless, semantic structures extracted from text-conditioned diffusion models (StableDiffusion) using our method are still relatively aligned with that of human annotators, rivaling those extracted from autoregressive Large Language Models while outperforming encoder-based language models such as BERT (Devlin et al., 2018).

| | STS-B | STS12 | STS13 | STS14 | STS15 | STS16 | SICK-R | Avg |
|---|---|---|---|---|---|---|---|---|
| *Contrastive-Trained Embedding Models* | | | | | | | | |
| CLIP-ViTL14[‡] (Radford et al., 2021) | 65.5 | 67.7 | 68.5 | 58.0 | 67.1 | 73.6 | 68.6 | $67.0 \pm 4.3$ |
| IS-BERT[†] (Zhang et al., 2020) | 56.8 | 69.2 | 61.2 | 75.2 | 70.2 | 69.2 | 64.3 | $66.6 \pm 5.7$ |
| SimCSE-BERT[†] (Gao et al., 2021) | 68.4 | 82.4 | 74.4 | 80.9 | 78.6 | 76.9 | 72.2 | $\mathbf{76.3} \pm 4.6$ |
| *Zero-Shot Encoder-based Models* | | | | | | | | |
| BERT-CLS[*] (Devlin et al., 2018) | 16.5 | 20.2 | 30.0 | 20.1 | 36.9 | 38.1 | 42.6 | $29.2 \pm 9.6$ |
| BERT-mean[*] (Devlin et al., 2018) | 45.4 | 38.8 | 58.0 | 58.0 | 63.1 | 61.1 | 58.4 | $54.8 \pm 8.3$ |
| BERT Large-mean[*] (Devlin et al., 2018) | 47.0 | 27.7 | 55.8 | 44.5 | 51.7 | 61.9 | 53.9 | $48.9 \pm 10.2$ |
| RoBERTa Large-mean[*] (Liu et al., 2019) | 50.6 | 33.6 | 57.2 | 45.7 | 63.0 | 61.2 | 58.4 | $52.8 \pm 9.6$ |
| ST5-Enc-mean (Large)[*] (Ni et al., 2021) | 56.3 | 28.0 | 52.6 | 41.4 | 61.3 | 63.6 | 59.5 | $51.8 \pm 11.9$ |
| ST5-Enc-mean (11B)[*] (Ni et al., 2021) | 62.8 | 35.0 | 60.2 | 47.6 | 66.4 | 70.6 | 63.6 | $\mathbf{58.0} \pm 11.5$ |
| *Autoregressive Models (Meanings as Trajectories (Liu et al., 2023))* | | | | | | | | |
| GPT-2[‡] | 55.2 | 39.9 | 42.6 | 30.5 | 52.4 | 62.7 | 62.0 | $49.3 \pm 11.2$ |
| GPT-2-XL[‡] | 62.1 | 43.6 | 54.8 | 37.7 | 61.3 | 68.2 | 68.4 | $56.5 \pm 11.1$ |
| Falcon-7B[‡] | 67.7 | 56.3 | 66.5 | 53.0 | 67.4 | 75.5 | 73.5 | $65.7 \pm 7.7$ |
| LLaMA-13B[‡] | 70.6 | 52.5 | 65.9 | 53.2 | 67.8 | 74.1 | 73.0 | $65.3 \pm 8.3$ |
| LLaMA-33B[‡] | 71.5 | 52.5 | 70.6 | 54.6 | 69.1 | 75.2 | 73.0 | $\mathbf{66.6} \pm 8.5$ |
| *Text-Conditioned Diffusion Models (StableDiffusion)* | | | | | | | | |
| Initial Timestep Prediction | 55.8 | 46.7 | 53.4 | 47.2 | 54.3 | 57.9 | 56.0 | $53.0 \pm 4.1$ |
| Final Timestep Prediction | 64.9 | 41.6 | 56.4 | 51.0 | 65.2 | 60.2 | 58.9 | $56.9 \pm 7.7$ |
| Direct Output Comparison | 57.0 | 44.7 | 45.6 | 43.3 | 58.5 | 56.2 | 53.5 | $51.3 \pm 6.0$ |
| Conjuring Semantic Similarity (KL-Div) | 69.1 | 56.9 | 60.6 | 59.5 | 71.5 | 65.7 | 64.8 | $64.0 \pm 4.9$ |
| Conjuring Semantic Similarity | 70.3 | 57.9 | 61.0 | 60.8 | 73.6 | 67.9 | 66.0 | $\mathbf{65.4} \pm 5.3$ |

In light of this, the experiments also suggest that our method can be an effective metric to quantify how well representation structures learnt by these text-encoders have been distilled to the resulting diffusion model. We explore this in the following section, where we study how faithfully learnt semantic relations between words transfer from text-encoders to the full diffusion model.

In Figure 2, we also qualitatively evaluate our method on measuring semantic similarity between various words. From the resulting pairwise similarity matrices, we can observe that words are closer in terms of their common hypernym class tend to cluster together, showing that our method can effectively capture word taxonomies. For instance, nouns that describe types of dogs cluster together, and nouns that describe types of marine animals similarly form another cluster. These two clusters are separate, in the sense that distances of words across these clusters are large than those within each cluster.

## 4.4 EMPIRICAL ANALYSIS

In this section, we run ablation studies using the STS-B dataset as a benchmark to explore the design space of our method, and improve its computational efficiency. We further analyze the failure modes of diffusion models through comparing semantic alignment of words belonging to different parts of speech.

**Prior over timestep distribution.** In our algorithm, we placed a uniform prior over timesteps $\{1, \ldots, T\}$. In Figure 3, we show that this simple choice works best when evaluated on alignment with human annotators on STS-B, as compared to other choices such as considering only a uniform

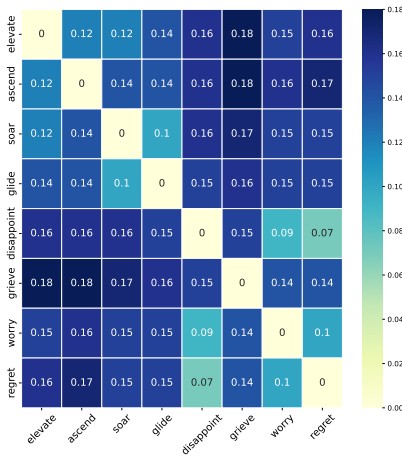

Figure 2: Qualitative evaluation of conjured semantic similarity. **(Left)** shows that nouns cluster based on shared hypernym classes: Dogs (puppy, poodle, dalmatian, pug) form a visible cluster in the top-left 4x4 block, while marine animals (whale, shark, dolphin, sealion) form another cluster in the bottom-right 4x4 block. **(Right)** shows that the same pattern holds for flying-related action verbs (elevate, ascend, soar, glide) v.s. negative stative verbs (disappoint, grieve, worry, regret).

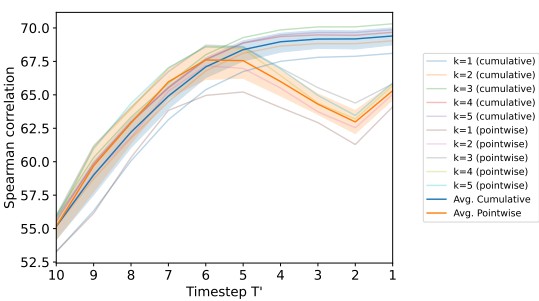 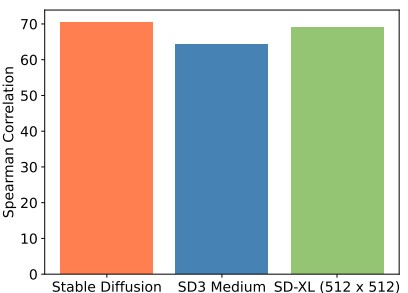

Figure 3: **(Left:)** We ablate over different choices of priors over timesteps – a uniform distribution over timesteps $\{T', \dots, T\}$ where $T' \leq T = 10$, represented by the blue line (cumulative), and the Direc Delta on any particular timestep $T' \in \{1, \dots, T\}$, represented by the orange line (pointwise). We show that a uniform prior over all timesteps gives the best results. The same plot also ablates over the number of Monte-Carlo samples, $k \in \{1, \dots, 5\}$, where we conclude that only few iterations are required for convergence. **(Right:)** We further ablate over different choices of diffusion models, and show that results remain relatively consistent across the tested choices.

prior over the subset $\{T', \dots, T\}$ (cumulative from $T'$ to $T$) where $T' \leq T$, or a Direc delta on any particular timestep $T' \in \{1, \dots, T\}$ (pointwise).

**Number of Monte-Carlo Steps.** The computational feasibility of our method depends on the number of Monte-Carlo steps (*i.e.* $k$ in Algorithm 1) required to produce a reliable approximate of the desired distance. We ablate over choices of $k \in \{1, \dots, 5\}$ in Figure 3 and show that the deviation of scores when evaluated on the STS-B dataset is small ($\pm 0.77$) across different choices of $k$. This finding is promising, as it implies that our method can be computationally efficient, requiring only a small number of Monte-Carlo iterations to converge.

**Choice of Stable Diffusion Model.** On the right of Figure 3, we also show that our results are relatively consistent across several versions of Stable Diffusion models, including Stable Diffusion XL and Stable Diffusion 3 Medium.

**Error Analysis.**   Here, we evaluate our method on word similarity datasets, categorized by the part of speech (POS) from which the words originate. We use RG65 (Rubenstein & Goodenough, 1965) and SimLex-999 (Hill et al., 2015) for our evaluation, which contains pairs of words and their semantic similarity score as annotated by humans. The former consists of nouns, while the latter includes adjectives, nouns, and verbs. Our analysis in Table 2 compares the representations learnt by the text encoder of the diffusion model to that of the model itself (extracted via our approach). Interestingly, we observe that while the semantic relations between nouns are largely preserved, the semantic relations between verbs and adjectives tend to deteriorate after learning the reverse diffusion process for image generation.

Table 2: We evaluate the spearman correlation between semantic similarity scores obtained via our method, and that labeled by human annotators. We use the text encoder bottleneck of the StableDiffusion model as the paragon. While the semantic properties of nouns are largely preserved from learning the diffusion process, this comes at the expense of semantics of adjectives and verbs.

|  | RG65 | SimLex | SimLex (Adj) | SimLex (Noun) | SimLex (Verb) |
|---|---|---|---|---|---|
| Paragon | 77.5 | 34.2 | 40.6 | 42.8 | 10.3 |
| StableDiffusion | 60.2 | 20.5 | 34.0 | 30.0 | -14.3 |

Table 3: Ablation on choices for $T$ on the STS-B dataset: We show that variance with respect to the choice of $T$ is small, allowing semantic distances to be computed efficiently by using lower values.

|  | $T = 5$ | $T = 10$ | $T = 15$ | $T = 20$ | $T = 30$ | $T = 50$ |
|---|---|---|---|---|---|---|
| Spearman Corr. | 70.1 | 70.3 | 70.1 | 68.9 | 70.2 | 69.5 |

## 5   DISCUSSION AND LIMITATIONS

Our method has several limitations. First, imageries might indeed not be sufficient to fully capture the meaning of certain expressions, such as mathematical abstractions (like 'imaginary numbers') and metaphysical concepts (like 'conscience'). Furthermore, many modern diffusion models use a pre-trained text-encoder to pre-process textual prompts. This means that representation structures obtained from the diffusion model outputs would be bottle-necked by those learnt by the text encoder. However, this limitation can be mitigated by the development of better encoders, such as those based on LLMs (BehnamGhader et al., 2024). Additionally, these encoder-based (vector) representations are often difficult to interpret, while our proposed method offers a way to visualize and interpret the learnt semantic similarities between textual expressions. Lastly, computation costs also remain a key limitation of our method, since it requires several inference passes through the diffusion model to compute a single semantic similarity score, mitigated only partially by the conclusions of our ablation study on required number of iterations.

Nevertheless, our method is the first to show that textual representations can be meaningfully compared for diffusion models by "grounding" them in the space of conjured images. We introduce the first method for evaluating semantic alignment of text-conditioned diffusion models. In particular, our work enables not only qualifying (via visual 'explanations'), but also quantifying the alignment of this resulting semantic space with that of human annotators. Our method also enable fine-grained analysis of the failure modes of existing diffusion models, pinpointing specific areas where they align poorly with human annotators. Our general framework of conjuring semantic similarity is applicable to the broader class of image generative models, and we leave the exploration of applications beyond diffusion models to future work.

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

# A    ADDITIONAL VISUALIZATIONS

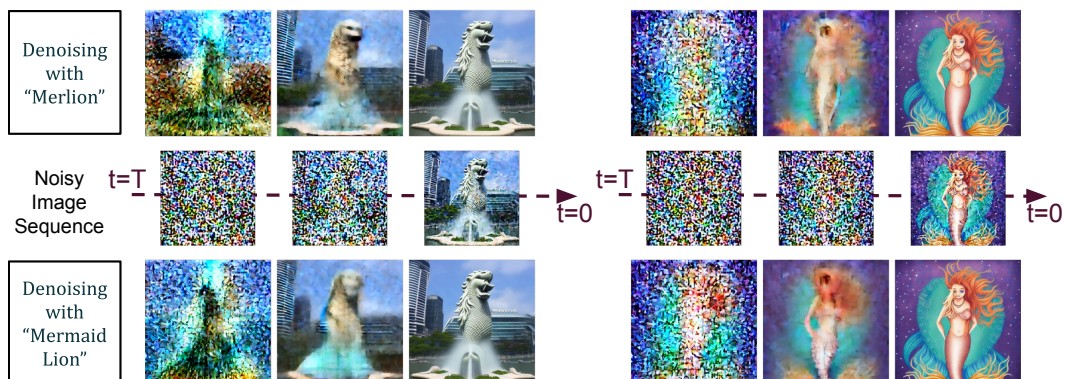

Figure 4: "Merlion" vs "Mermaid Lion": While both prompts express compositions of the same set of objects, the model associates different meanings with "Merlion" as opposed to "Mermaid + Lion", where the former is associated to the mascot of Singapore, while the latter is a mermaid with hair resembling a lion's mane.

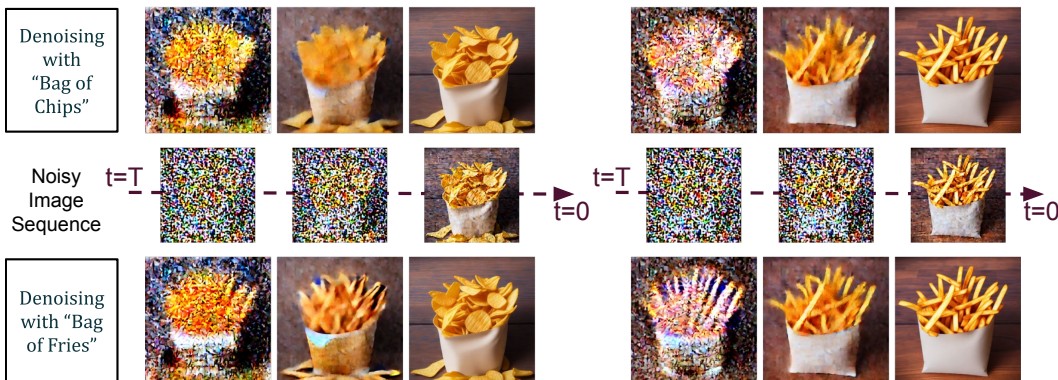

Figure 5: "Bag of Chips" vs "Bag of Fries": The interpretation of "chips" depends on cultural background (US vs UK), but the interpretation of "fries" is relatively non-ambiguous. Interestingly, this observation can be visualized when computing semantic similarity with our method. We see that on the left of the figure (second image column), the model attempts to convert a picture of chips (US) into fries by changing the rounded textures into sharper rectangular ones, when denoised with "Bag of Fries". On the other hand, pictures of fries still remain relatively identifiable as fries (fifth image column) when denoised using "Bag of Chips".

