# OpenReview forum: "Conjuring Semantic Similarity"
_ICLR.cc/2025/Conference — Submitted to ICLR 2025_

### Official Review · Reviewer_WVfY · 2024-11-01

**Soundness:** 2
**Presentation:** 3
**Contribution:** 2
**Rating:** 5
**Confidence:** 3

**Summary:**

This paper proposes a novel method for calculating the similarity between two pieces of text that is visually grounded. For each sentence, the authors use the sentence to guide a text-conditioned diffusion model to remove the noise and generate a new image given some initial noisy image.
Each sentence guides the diffusion model to generate intermediate denoised images with a distinct distribution. To compare two sentences, the authors define the distance between the two sentences as the distance between the corresponding distributions of the intermediate denoised images.

**Strengths:**

* The paper is well-written and easy to read.
* The idea of a visually grounded similarity metric between sentences is very interesting and could be a useful addition to the community’s toolbox for measuring different aspects of semantic similarity.
* The authors provide a theoretical understanding and justification for the introduced distance metric.

**Weaknesses:**

My main concern with this paper is the lack of a clear contribution that is well-motivated and also well-supported by the presented empirical evidence. In the last paragraph of section 1, the authors claim three separate contributions. Here are my concerns about each claimed contribution:

First contribution (Lines 82-83): “we propose an approach for evaluating semantic similarity between text expressions that is grounded in the space of visual images.”

Although I agree that a novel notion of similarity metric is interesting, the paper lacks a solid argument to support why it is needed or useful. Some of the unanswered questions are:
* Why do we need a new similarity metric? In paragraph 2, the authors provide some arguments about why comparing images is easier for humans than comparing text due to language and knowledge differences. But, that argument does not hold for models since the similarity between texts is often calculated by a single model. So, there is no difference in knowledge or judgment between models. Moreover, even if this limitation is actually valid for text similarity measures based on dense vectors, it also applies to the proposed similarity metric. After all, just like there is model that is creating the dense vector, there is a model that is doing the denoising and image generation. So, there is still a model involved with all the specific and often unknown limitations and biases that come with any model.
* As the authors mentioned in Line 312, these generative models rely on some encoder to encode the text. So, all the limitations of the encoder model (i.e., cosine similarity between dense vectors) also apply to the proposed similarity metric. So, why should we add the extra level of complexity?
* Even beyond the motivation, the experimental results do not suggest that the new similarity metric is consistently better than the vector-based methods. The most interesting comparison is the performance of the proposed similarity metric compared to the performance of the text encoder that is used in the stable diffusion generative model, which is clip-vit-14. Comparing these two, it seems like in just three out of the seven datasets, the new similarity metric performs better, which is less than half the times.
* In general, to show the superior performance of the proposed similarity metric, more datasets that cover more diverse domains are needed.

Second contribution (Lines 84-85): “Our method has a unique advantage over traditional language-based methods that, in addition to providing a numerical score, it also provides a visual expression, or ‘explanation’, for comparison, enabling better interpretability of the learnt representations”:

I completely agree with this advantage of the proposed method, and I think it might be the exact case where their method shines. But, other than three figures that show qualitatively what this interpretation might look like, there is no other discussion or experiment on this.
To claim this as the main contribution, the authors should provide extensive experiments and discussion on the types of explanation that their method provides in different use cases, discuss how these explanations can be useful for the community, and how their method compares to existing interpretability methods.


Third contribution (Lines 86-87): “our method is the first to enable quantifying the alignment of semantic representations learnt by diffusion models compared to that of humans, which can open up new avenues for the evaluation text-conditioned diffusion models”

Again, I completely agree that this can be a strength of the proposed method. But, similar to my previous point, to claim this as the main contribution, the authors should evaluate several diffusion models, discuss helpful insights that their metric provides, compare their evaluation results with previous evaluation metrics, and explain their differences, advantages, and weaknesses.

---

I also have an issue with the lack of a baseline to measure the merits of the proposed similarity metric. This paper proposes two different concepts. One is the notion of text similarity through images, and the second is the proposed similarity metric to accomplish this. Assuming that the authors provide adequate motivation and support for the importance of the notion of text similarity through evoked images, they should also prove that the complexity of their similarity metric is justified. For example, what if I just generate the final fully denoised images using each sentence and then measure the similarity of the generated images using something like CLIP or DINO embeddings?

Finally, I think the limitations that the authors mention in Section 5 deserve more than a one-sentence acknowledgment. For example, the authors mention that their method is computationally expensive. It would be helpful for the community to know the exact computational resources used for the experiments.
The authors also mention the limitations that are caused by the use of text encoders such as CLIP by diffusion models. It is important to explore and analyze these limitations and also show the merits of the proposed method compared to just using the outputs of the text encoder with cosine similarity in the first place.

**Questions:**

Please see the weaknesses.

---

> ### Author Response · Authors · 2024-11-17
>
> We thank the reviewer for their valuable feedback and suggestions.
>
> > Why do we need a new similarity metric?
>
> This is because no existing metric is satisfactory (Stein et al., 2024), thus the evaluation of text-conditioned diffusion models remains an active area of investigation. In fact, no metric even exists to evaluate semantic similarity and alignment in text-conditioned diffusion models. Our work is the first to do so, by adding a new dimension (images) to the problem that had never been considered before. Having this also opens up avenues for evaluating semantic alignment with human annotators, as demonstrated in our experiments.
>
> > In paragraph 2, the authors provide some arguments about why comparing images is easier for humans than comparing text due to language and knowledge differences. But, that argument does not hold for models since the similarity between texts is often calculated by a single model. So, there is no difference in knowledge or judgment between models.
>
> Indeed semantic spaces are uniquely defined for each individual model. However, this is a separate point from what we wished to convey in the last line of paragraph 2 -- that for humans, unlike images, comparing text requires a stronger shared knowledge base (for instance of the language in which the text is written). Nevertheless, we admit this sentence is debatable, and are willing to remove it from the final revision.
>
>
> > Moreover, even if this limitation is actually valid for text similarity measures based on dense vectors, it also applies to the proposed similarity metric. After all, just like there is model that is creating the dense vector, there is a model that is doing the denoising and image generation. So, there is still a model involved with all the specific and often unknown limitations and biases that come with any model.
>
> The fact that semantic similarity is model/human-specific is precisely the motivation of our work -- which is to be able to quantify how semantically aligned different (image generation) models are with that of human annotators. We elaborate further in our response to W2 of Reviewer rCYr.
>
> > As the authors mentioned in Line 312, these generative models rely on some encoder to encode the text. So, all the limitations of the encoder model (i.e., cosine similarity between dense vectors) also apply to the proposed similarity metric. So, why should we add the extra level of complexity?
>
> The "extra level of complexity" is actually a feature of our method: We perform comparison in the semantic space learnt by the generative model, rather than the text encoder itself. While, as mentioned in L312 and rightfully pointed out by the reviewer, the representations learnt are bottlenecked by the text encoder, this does not mean that all the information captured by the encoder is faithfully transferred to the generative model. We refer to the example given in the response to Reviewer rCYr -- a model trained to generate the same image for all text inputs can have a well-aligned text-encoder, but it would be inappropriate to conclude that this generative model is semantically well-aligned with humans.
>
> > Even beyond the motivation, the experimental results do not suggest that the new similarity metric is consistently better than the vector-based methods. The most interesting comparison is the performance of the proposed similarity metric compared to the performance of the text encoder that is used in the stable diffusion generative model, which is clip-vit-14. Comparing these two, it seems like in just three out of the seven datasets, the new similarity metric performs better, which is less than half the times.
>
> > I also have an issue with the lack of a baseline to measure the merits of the proposed similarity metric. This paper proposes two different concepts. One is the notion of text similarity through images, and the second is the proposed similarity metric to accomplish this.
>
>
> The vector-based methods are included as points of reference, rather than for purposes of direct comparison. Not only are they used to evaluate completely different models, but none of them can be used to capture semantic similarity for these text-conditioned image generative models.
>
> What can be directly compared, however, are the new baselines included in the revision. We refer the reviewer to the updated results, where we show that our method outperforms **all** comparable baselines across all datasets.

---

> > ### Author Response · Authors · 2024-11-17
> >
> > > In general, to show the superior performance of the proposed similarity metric, more datasets that cover more diverse domains are needed.
> >
> > We have updated our latest revision with new experiments on RG65 and SimLex-999, alongside the other domains we evaluate in our paper -  WordNet, STS, and SICK-R.
> >
> >
> > > I completely agree with this advantage of the proposed method, and I think it might be the exact case where their method shines. But, other than three figures that show qualitatively what this interpretation might look like, there is no other discussion or experiment on this. To claim this as the main contribution, the authors should provide extensive experiments and discussion on the types of explanation that their method provides in different use cases, discuss how these explanations can be useful for the community, and how their method compares to existing interpretability methods.
> >
> > To address the reviewer's concern, we have included Table 2 in our latest revision to showcase how our method can be used to interpret what classes of semantic relations are captured by the diffusion model.
> >
> > While (visual) interpretability is an additional benefit of our method, the main goal of our paper is to introduce a novel first method to evaluate semantic similarity via image generative models. While we provide supporting experiments, we leave further extensive exploration on different possible aspects of interpretability to follow-up work.
> >
> >
> > >  But, similar to my previous point, to claim this as the main contribution, the authors should evaluate several diffusion models, discuss helpful insights that their metric provides, compare their evaluation results with previous evaluation metrics, and explain their differences, advantages, and weaknesses.
> >
> > Our method is the first to allow for quantifying semantic alignment of diffusion models, hence orthogonal to other metrics for measuring quality and diversity, which we discuss in L138-150. We have updated our revision to make this distinction clearer.
> >
> >
> > >  Assuming that the authors provide adequate motivation and support for the importance of the notion of text similarity through evoked images, they should also prove that the complexity of their similarity metric is justified. For example, what if I just generate the final fully denoised images using each sentence and then measure the similarity of the generated images using something like CLIP or DINO embeddings?
> >
> > This specific suggestion of the reviewer requires introducing an additional model into the pipeline, which actually increases the level of complexity while reducing the ability to faithfully evaluate representations learnt by the diffusion model itself. However, we have incorporated an alternate version of the reviewer's proposed baseline in our revision in Table 1, under Direct Output Comparison, and thank the reviewer for their suggestion.
> >
> >
> > > Finally, I think the limitations that the authors mention in Section 5 deserve more than a one-sentence acknowledgment. For example, the authors mention that their method is computationally expensive. It would be helpful for the community to know the exact computational resources used for the experiments.
> >
> > We have updated Sec 4.1 to include computational details, thank you for pointing this out.
> >
> > > The authors also mention the limitations that are caused by the use of text encoders such as CLIP by diffusion models. It is important to explore and analyze these limitations and also show the merits of the proposed method compared to just using the outputs of the text encoder with cosine similarity in the first place.
> >
> > We have modified Sec 4 to incorporate this discussion.

---

> > > ### Comment · Reviewer_WVfY · 2024-11-27
> > >
> > > Thank you for your response! I have updated final score!

---

> > > > ### Author Response · Authors · 2024-11-27
> > > >
> > > > Thank you for your response, and for updating your score. We truly appreciate your detailed review, which has been invaluable in improving our work. We would greatly appreciate it if you could share any further insights or suggestions. Otherwise if you feel we have satisfactorily addressed your concerns, we kindly ask if you could consider increasing your score to support our paper's acceptance. Thanks!

---

> > > > > ### Author Response · Authors · 2024-12-03
> > > > >
> > > > > Given the approaching rebuttal deadline, we wish to follow up our response to your feedback. We hope that our clarifications, revised paper draft, and additional experiments have addressed the areas of weakness you identified. If there are remaining questions or further points of weaknesses, we kindly ask if you can detail them so we may address them. Thank you!

---

### Official Review · Reviewer_tXAe · 2024-11-02

**Soundness:** 2
**Presentation:** 3
**Contribution:** 2
**Rating:** 6
**Confidence:** 3

**Summary:**

The paper discusses the measurement of semantic similarity at the concept level. The authors propose an interesting method to measure semantic similarity via the distribution of the image with guidance from paired different textual expressions. On various STS tasks, the authors show comparable performance between the proposed methods and various textual or multi-modal semantic similarity measurements.

**Strengths:**

1.	This paper proposes an interesting perspective linking text and diffusion models. The idea of contextual and concept-level similarity is natural and convincing.
2.	This paper is well-written and easy to follow.

**Weaknesses:**

1.	The empirical experiments are too weak. On the one hand, the performance is not as good as the previous methods. The baselines listed are slightly outdated. It would be interesting if OpenAI ada embeddings, Llama 3.1, and Gemma can be compared. Surpassing BERT-CLS is not very convincing. On the other hand, STS may not be the best arena for the novel method the authors proposed, since there are a lot of examples related to paraphrasing, instead of conceptual relevance. It would be great if the authors study a specific slice that is more relevant to the idea, e.g., examples with a not-complex scene and the difference mainly comes from the concepts. Knowledge graphs (e.g., COMET, Atomic) or WordNet depth distance may provide better comparative performance or qualitative examples.
2.	This paper can be linked with more recent advances in this area in NLP, e.g., C-STS (https://arxiv.org/abs/2305.15093) discusses the conditional semantic textual similarity in addition to STS; Instructor (https://github.com/xlang-ai/instructor-embedding) and PIR (https://arxiv.org/abs/2405.02714) discuss the changes in semantic embeddings under different instructions. The related work and experiment suite can help further improve the draft in discussing the semantic similarity.
3.	It would be good if further content is added to the paper especially since there are 7.5 pages in the current draft. For example, the authors can include experiments on potential performance improvement over various downstream tasks: Z-LaVI (https://arxiv.org/abs/2210.12261), for example, explores how visual imagination helps improve model performance on tasks such as word sense disambiguation in a zero-shot manner. Similar tasks can be included in the draft.

**Questions:**

1.	Is there a specific set of STS that you found the proposed method works?

---

> ### Author Response · Authors · 2024-11-17
>
> We thank the reviewer for their detailed feedback and their interesting ideas on extending our work.
>
> > The empirical experiments are too weak. On the one hand, the performance is not as good as the previous methods. The baselines listed are slightly outdated. It would be interesting if OpenAI ada embeddings, Llama 3.1, and Gemma can be compared. Surpassing BERT-CLS is not very convincing.
>
> Our method is in fact the first work that evaluates this aspect of semantic alignment for text-conditioned diffusion models. Consequently, there are no proper baselines, and comparison with text-based methods in Table 1 is somewhat reductive as we aim to explore a different dimension - the visual one. Indeed there is much more evaluation method to be done to probe the value of visual representations for comparing textual semantics, which is well beyond of the scope of a single conference paper that introduces the concept.
>
> However, what can be considered comparable baselines are the new methods we included in Table 1, where we tested other possible approaches to extract semantic similarity from diffusion models. Our approach, which is theoretically grounded by considering the JSD between SDEs induced by various textual expressions, convincingly outperforms these baselines across all datasets.
>
> > On the other hand, STS may not be the best arena for the novel method the authors proposed, since there are a lot of examples related to paraphrasing, instead of conceptual relevance. It would be great if the authors study a specific slice that is more relevant to the idea, e.g., examples with a not-complex scene and the difference mainly comes from the concepts. Knowledge graphs (e.g., COMET, Atomic) or WordNet depth distance may provide better comparative performance or qualitative examples.
>
> While we included some WordNet-based evaluations in Figure 2, we have added a stronger analysis at the end of Sec 4.4 studying semantic similarity between words, which can be stratified based on the parts of speech from which they originate. We thank the reviewer for their excellent suggestion!
>
> > This paper can be linked with more recent advances in this area in NLP, e.g., C-STS ([https://arxiv.org/abs/2305.15093](https://arxiv.org/abs/2305.15093)) discusses the conditional semantic textual similarity in addition to STS; Instructor ([https://github.com/xlang-ai/instructor-embedding](https://github.com/xlang-ai/instructor-embedding)) and PIR ([https://arxiv.org/abs/2405.02714](https://arxiv.org/abs/2405.02714)) discuss the changes in semantic embeddings under different instructions. The related work and experiment suite can help further improve the draft in discussing the semantic similarity.
>
> We thank the reviewer for the references, and will add a discussion to the paper on how our paper can extend to these areas after perusing them carefully.
>
> > It would be good if further content is added to the paper especially since there are 7.5 pages in the current draft. For example, the authors can include experiments on potential performance improvement over various downstream tasks: Z-LaVI ([https://arxiv.org/abs/2210.12261](https://arxiv.org/abs/2210.12261)), for example, explores how visual imagination helps improve model performance on tasks such as word sense disambiguation in a zero-shot manner. Similar tasks can be included in the draft.
>
> Indeed our work opens up many possibilities for evaluating the semantic alignment of image generative models in future work. We hope that our new experiments and analysis should address the reviewers concern regarding paper content, while the reviewer’s valid and interesting idea will be explored in future work.
>
>
> > Is there a specific set of STS that you found the proposed method works?
>
> A low score implies low level of semantic alignment, while high scores simply imply high levels of semantic alignment of the specific diffusion model under evaluation. Compared, however, to naive diffusion baselines for extracting semantic similarity, we have updated Table 1 to show that our method greatly outperforms baselines on **all** sets of STS and SICK-R. The newly added Table 2 also analyzes alignment on word pairs, stratified across different parts of speech. Interestingly, we found that semantic relations between nouns are largely preserved across the diffusion training process, but this comes at the expense of adjectives and verbs.

---

> > ### Comment · Reviewer_tXAe · 2024-11-25
> > **Thank you for your reply**
> >
> > The reviewer sincerely appreciates your reply and the update of the draft. I have updated the score accordingly. The novelty of the work is well acknowledged. Showing application on the downstream tasks would further improve the quality of the paper.

---

> > > ### Author Response · Authors · 2024-11-27
> > >
> > > Thank you for your response and supporting the acceptance of our paper! We are very grateful for your interesting and valid suggestions on potential future work, and are excitedly looking forward to new research directions and applications that our work opens up.

---

### Official Review · Reviewer_yfJv · 2024-11-02

**Soundness:** 3
**Presentation:** 3
**Contribution:** 2
**Rating:** 6
**Confidence:** 3

**Summary:**

The authors propose to use the distance between image distributions generated by a diffusion model conditioned on two text sequences as a novel measure of semantic similarity.   They specifically propose using the Jensen-Shannon divergence between the reverse-time diffusion stochastic differential equations (SDEs) induced by two text sequences which can be directly computed via Monte-Carlo sampling.

Empirically, while not performing as well as embedding models trained specifically for semantic comparison tasks  ( CLIP, SimCSE-BERT, etc ), the author's approach outperforms zero-short encoder models and aligns well with human-annotated scores from the Semantic Textual Similarity and Sentences involving Compositional Knowledge benchmarks.

The authors provide some findings from ablations over the choice of prior distributions (uniform vs dirac), number of monte carlo steps and the choice of diffusion model as well.

**Strengths:**

The work proposes a novel method for assessing semantic similarity between two text sequences by comparing the distance between image distributions generated over time by diffusion models conditioned on them.

The authors show the method does well on human annotation semantic similarity baselines.

The method offers a new possible avenue for the evaluation of text-conditioned generative models which allows some interpretability of representation similarities for text conditioned image generation models.   While its not the most performative method, it seems like one that could be expanded/built upon both to improve its performance, but also possibly as a way of guiding/improving diffusion model training.

**Weaknesses:**

While the authors cited weaknesses with the approach, the authors could do a better job motivating possible applications opened up by their method.

The paper qualitative experiment is a little shallow. Expanding it and doing error analysis ( where does the method perform well/poorly ) on results would help add more clarity into the method.

The interpretability angle of the paper is also a little lacking and under explained.  Outside of the few examples given (one in the paper and two in the appendix ), is there anyway to automate interpretability results or use them for error analysis in a useful way.

The paper could have done an ablation compared their symmetric JSD approach to just using KL-Divergence to show the boost obtained.

The last line of the 2nd paragraph ( about pixel values not depending on distant knowledge or cultural background ) seems debatable/unnecessary since a text passage (about democracy, celebration, a feast, etc ) could be portrayed visually in different ways depending very much on knowledge/cultural background while still referring to the same semantic concept

**Questions:**

1) on line 238, "denoising with either y1 and y2" is a little confusing in that "either" seems to imply an exclusive OR while "and", Algo 1 and the definition d_ours(y1,y2) show the need for both?  I could be mistaken, but I think the later is the case, in which case you should change "either" to be "both"

2) The line starting at 257 was a little unclear to me.  Specifically "we note that they do not usually correspond exactly" ?

3) nit: 313 "is" -> "are"

---

> ### Author Response · Authors · 2024-11-17
>
> We are very grateful for the reviewer's constructive feedback, and incorporated their suggestions into our revision. We address them in detail below:
>
> > While the authors cited weaknesses with the approach, the authors could do a better job motivating possible applications opened up by their method.
>
> We have modified our paper, especially the last part of Sec 5, to incorporate this suggestion, thank you.
>
> > The paper qualitative experiment is a little shallow. Expanding it and doing error analysis ( where does the method perform well/poorly ) on results would help add more clarity into the method.
>
>
> To address both the qualitative and quantitative aspect, we have incorporated a range of new experiments. We refer the reviewer to the overall comment for the full changelog.
>
> > The paper could have done an ablation compared their symmetric JSD approach to just using KL-Divergence to show the boost obtained.
>
> We thank the reviewer for the suggestion and incorporated this into Table 1. However, we stress our goal is not to beat the benchmarks for semantic similarity, but to explore an alternate dimension, not (yet) well represented in existing methods, in the context of image generative models. While KL-divergence is also a valid method to do so, it is fundamentally asymmetric and fails to capture the fact that semantic similarity (at least for humans) is largely a symmetric property.
>
>
> > The interpretability angle of the paper is also a little lacking and under explained. Outside of the few examples given (one in the paper and two in the appendix ), is there anyway to automate interpretability results or use them for error analysis in a useful way.
>
> Our method produces both a semantic similarity score, as well as an interpretable explanation in the form of generated images. The score can be used for analyzing alignment with human annotations in a quantitative manner. The justification for this score can then be visualized if desired, as shown in the qualitative evaluations.
>
> One possible way for error analysis would be leveraging additional meta-data in the evaluation data. We have added an experiment on this on the SimLex-999 dataset to quantify errors across different POS (part of speech) classes in Sec 4.4, and thank the reviewer for their valuable suggestion!
>
> > The last line of the 2nd paragraph ( about pixel values not depending on distant knowledge or cultural background ) seems debatable/unnecessary since a text passage (about democracy, celebration, a feast, etc ) could be portrayed visually in different ways depending very much on knowledge/cultural background while still referring to the same semantic concept
>
> We wished to convey the point that text needs to be interpreted before it can be semantically, or even synthetically, compared, while images require significantly less processing to compare directly. However, we do acknowledge that this is highly debatable, and could remove it in the final revision.
>
>
> > on line 238, "denoising with either y1 and y2" is a little confusing in that "either" seems to imply an exclusive OR while "and", Algo 1 and the definition d_ours(y1,y2) show the need for both? I could be mistaken, but I think the later is the case, in which case you should change "either" to be "both"
>
> Indeed the latter is the case. "Either" was used to refer to what is being done in a *single* Monte-Carlo step, but we realized that using "both" in the context presented in L238 might be more appropriate. We have corrected this in the revision, thank you for catching this.
>
> > The line starting at 257 was a little unclear to me. Specifically "we note that they do not usually correspond exactly" ?
> > nit: 313 "is" -> "are"
>
> Fixed, thanks!

---

> > ### Comment · Reviewer_yfJv · 2024-11-25
> > **author rebutal reply**
> >
> > I appreciate the baseline updates, ablations and minor fixes address.  While I still think the quantitative interpretability angle needs to be expanded on more ( to include both more examples like the newly made figure 1 which is much clearer ) to show how these visuals could be used.
> >
> > The newly added experiment and results table 2 are a little unclear to me ( as it lacks the metric being shared in the table and context about the datasets/tasks ).  I feel clarifying this some in addition to giving more concrete visual explanation use cases for the method will make this a further stronger paper and have updated my score assuming those small additions/changes can be made.
> >
> > Nit to fix: line 473 "but also quantifying but also quantifying ..."

---

> > > ### Author Response · Authors · 2024-11-27
> > >
> > > We have updated our revision to incorporate your feedback and provide additional context about the evaluation metric and dataset description. We will also prepare a few additional qualitative examples in the style of Fig. 1, and Appendix A in the meantime. We are grateful for your detailed and helpful suggestions, and for supporting the acceptance of our work. Thank you!

---

### Official Review · Reviewer_rCYr · 2024-11-04

**Soundness:** 3
**Presentation:** 3
**Contribution:** 2
**Rating:** 6
**Confidence:** 3

**Summary:**

The paper proposes a novel similarity measurement–textual similarity based on the imagery the texts evoke. They propose learning this similarity by computing the Jensen-Shannon divergence between diffusion processes conditioned on the two compared prompts, using Monte-Carlo sampling. The method is then compared to existing similarity measurements on the STS benchmarks and is ablated.

**Strengths:**

**S1:** The idea of learning semantic textual similarity through the images such expressions evoke is creative, original, and intriguing.

**S2:** Related Work provides a detailed and relevant account of existing work in the subject.

**Weaknesses:**

**W1:** The premise of the paper–obtaining a similarity measurement based on the imagery texts evoke–is unique and interesting, but I don’t understand what use-case is it tailored to address. If the use-case is no different than measuring similarities in text-only environments, I don’t see why it is preferable over methods that are inherently text-only, are easier to scale, and are equipped to represent abstract notions, which are difficult to visualize. I find that conclusion is supported in section 4.2 and table 1, too.

The paper will be improved if such motivation will be clarified, with an experiment to demonstrate this use-case.

**W2:** Isn’t the method hindered by whatever the text encoder does not represent well, or what the diffusion process did not accurately learn? Seeing as these are inherent components, this seems like a significant drawback, which effectively nullifies possible advantages of this method. You touch on this matter in lines 312–317 and section 5. I would appreciate a clarification here.

**Questions:**

**Questions:**
Is it possible to detect poor representations automatically in the underlying text encoder using this similarity measure?


**Other:**
I had a hard time understanding Figure 1, consider redesigning it or breaking it down to two figures

---

> ### Author Response · Authors · 2024-11-17
>
> We thank the reviewer for their helpful feedback, and address concerns below:
>
> >**W1:** The premise of the paper–obtaining a similarity measurement based on the imagery texts evoke–is unique and interesting, but I don’t understand what use-case is it tailored to address. If the use-case is no different than measuring similarities in text-only environments, I don’t see why it is preferable over methods that are inherently text-only, are easier to scale, and are equipped to represent abstract notions, which are difficult to visualize. I find that conclusion is supported in section 4.2 and table 1, too. The paper will be improved if such motivation will be clarified, with an experiment to demonstrate this use-case.
>
> Our work aims to explore a different dimension of semantic similarity based on image generative models. There are (at least) two use cases: One is self-evident, which is to provide alternate methods to measure semantic similarity in text, by opening a new modality of analysis - which is visual. The other is to provide methods to evaluate and compare text-conditioned diffusion-based image generation models - although this is largely open for investigation and well beyond the scope of a single conference paper whose goal is to introduce the concept. Importantly, none of the existing text-only methods pertain to either of these use-cases.
>
> However, what can be empirically tested are baseline methods for extracting semantic similarity targeted towards image diffusion models. While we are the first work to formalize this notion of semantic similarity/alignment for such models, we introduced several new comparable baselines in our revision (see overall comment) to address the reviewer's concern. We have also further modified Sec 4.3 in our paper to incorporate this discussion on use-cases.
>
>
> > **W2:** Isn’t the method hindered by whatever the text encoder does not represent well, or what the diffusion process did not accurately learn? Seeing as these are inherent components, this seems like a significant drawback, which effectively nullifies possible advantages of this method. You touch on this matter in lines 312–317 and section 5. I would appreciate a clarification here.
>
>
> Conceptual or semantic similarity is intrinsically subjective or model-dependent, as it can be proven that there exist no canonical notion of semantic information, and therefore semantic similarity (shared information) -- see Achille et al., 2024. The fact that the method can reflect what is learnt by both the text encoder **and** the diffusion process is precisely our advantage. While the semantic alignment of modern diffusion models are indeed "upper-bounded" by representation structures captured by text-encoders, this does not imply equality. For instance, one can easily train a model to map all encoded text to a white background. Even if the original text encodings are highly informative, it would be inappropriate to claim that the resulting model is semantically well-aligned. We also refer to our new experiments in Sec 4.4, which demonstrates a case where diffusion models preserve semantic relations between only certain categories of words.
>
> This also relates to the reviewer's next question:
>
> > Is it possible to detect poor representations automatically in the underlying text encoder using this similarity measure?
>
> It is possible, but the result will be conflated with the representations learnt from the diffusion training process (it would be better to independently evaluate the text encoder unit on its own). However, what is enabled by our method is the detection of how well semantic relations captured by the text encoder are transferred to the trained diffusion model, as our experiments in Sec 4.4 demonstrates.
>
> > I had a hard time understanding Figure 1, consider redesigning it or breaking it down to two figures
>
> We have redesigned Figure 1 to incorporate the reviewer's suggestion, thank you for the suggestion!

---

> > ### Author Response · Authors · 2024-12-03
> >
> > Given the approaching rebuttal deadline, we wish to follow up on our response to your insightful feedback. We hope that our responses, revised paper draft, and additional experiments have addressed the areas of weakness you identified. We would be grateful if you could consider providing stronger support for our paper's acceptance, and thank you for your very valuable feedback and suggestions!

---

### Author Response · Authors · 2024-11-17
**Paper Submission Updated**

We are happy to hear that our work is well-received by the reviewers with regards to its “creativity”, “originality”, “novelty” (rCYr, yfJv), and that our idea of “visually grounded semantic similarity” being an “interesting perspective linking text and diffusion models” is “natural and convincing” while backed up by “theoretical understanding and justification” (tXAe, WVfY). We are also heartened to see reviewers actively proposing novel directions of research that our work has opened up, including “a way of guiding/improving diffusion model training” (yfJv), “exploring how visual imagination helps improve model performance” (tXAe), and as a “useful addition to the community’s toolbox for measuring different aspects of semantic similarity” (WVfY).

We greatly appreciate the reviewers' valuable feedback, and have incorporated several of their suggestions in our paper revision. Our paper is the first work to define a (theoretically grounded) notion of semantic similarity for text-conditioned diffusion models, which we show can be used for evaluating semantic alignment of such models with human annotators. Nonetheless, we have included several comparable baselines in our new revision, where we show that our approach indeed outperforms all other methods used for quantifying semantic alignment in diffusion models.

We have also expanded our evaluation to include word similarity datasets, stratified based on the parts of speech from which the words originate. This allows for finer-grained analysis and interpretation of the semantic relations captured by diffusion models, and we refer to our revised draft for the full results and discussion.

To summarize, our revision includes, but is not limited to, the following changes:

- Added comparisons to four new baselines in Table 1 - Initial Timestep Prediction, Final Timestep Prediction, Direct Output Comparison, and a version of our method based on KL-Divergence, where we show that our method outperforms all baselines across all datasets.
- Added experiments on RG65 and SimLex-999 datasets in Table 2.
- Added an analysis on failure modes of diffusion models, with respect to preserving semantic relations learnt by their text-encoders, in Sec 4.4.
- Added Sec 4.2 to motivate baselines.
- Expanded contributions of our method in Sec 5.
- Modified Figure 1
- Other changes discussed in individual responses to reviewers

---

### Meta-Review · Area_Chair_JkxJ · 2024-12-19

**Metareview:**

The paper proposes a novel measurement of semantic similarity based on the imagery of the texts. The authors propose using Jensen-Shannon divergence (JSD) between the diffusion  process conditioned on the input  text, via Monte-Carlo sampling.

The reviewers agreed the proposed approach is novel and highlight the paper is well-written and easy to follow. The idea of a visually grounded similarity metric between sentences is interesting and has its potential to be expanded.

The major concerns from the reviewers are:
1. Lack of the practical usage of the proposed approach, it is not better than existing textural similar models. And it could be slower with the diffusion process.
2. Deeper analysis / more evidence for better understand the proposed approach including incorporating with other text based approaches, the interpretability, and limitation.

After discussion, the reviewers agree this is an interesting submission but can be improved significantly.

**Additional Comments On Reviewer Discussion:**

Please see the major concerns from the meta review. I think they are valid. The authors added more experiment results like word similarity datasets, stratified based on the parts of speech from which the words originate. But reviewers are still not convinced.

---

### Decision · Program_Chairs · 2025-01-22

Reject